# Crystallization Mechanism and Optical Properties of Antimony-Germanate-Silicate Glass-Ceramic Doped with Europium Ions

**DOI:** 10.3390/ma15113797

**Published:** 2022-05-26

**Authors:** Piotr Golonko, Karolina Sadowska, Tomasz Ragiń, Marcin Kochanowicz, Piotr Miluski, Jan Dorosz, Marta Kuwik, Wojciech Pisarski, Joanna Pisarska, Magdalena Leśniak, Dominik Dorosz, Jacek Żmojda

**Affiliations:** 1Faculty of Electrical Engineering, Bialystok University of Technology, 45D Wiejska Street, 15-351 Bialystok, Poland; p.golonko@doktoranci.pb.edu.pl (P.G.); k.sadowska@doktoranci.pb.edu.pl (K.S.); tomasz.ragin@pb.edu.pl (T.R.); m.kochanowicz@pb.edu.pl (M.K.); p.miluski@pb.edu.pl (P.M.); doroszjan@pb.edu.pl (J.D.); 2Institute of Chemistry, University of Silesia, 9 Szkolna Street, 40-007 Katowice, Poland; marta.kuwik88@gmail.com (M.K.); wojciech.pisarski@us.edu.pl (W.P.); joanna.pisarska@us.edu.pl (J.P.); 3Faculty of Materials Science and Ceramics, AGH University of Science and Technology, 30 Mickiewicza Av., 30-059 Krakow, Poland; mlesniak@agh.edu.pl (M.L.); ddorosz@agh.edu.pl (D.D.)

**Keywords:** active glass-ceramic, luminescent properties, nucleation, precursor of crystallization, antimony-germanate glass, Eu^3+^ ions, asymmetry ratio

## Abstract

Glass-ceramic is semi-novel material with many applications, but it is still problematic in obtaining fibers. This paper aims to develop a new glass-ceramic material that is a compromise between crystallization, thermal stability, and optical properties required for optical fiber technology. This compromise is made possible by an alternative method with a controlled crystallization process and a suitable choice of the chemical composition of the core material. In this way, the annealing process is eliminated, and the core material adopts a glass-ceramic character with high transparency directly in the drawing process. In the experiment, low phonon antimony-germanate-silicate glass (SGS) doped with Eu^3+^ ions and different concentrations of P_2_O_5_ were fabricated. The glass material crystallized during the cooling process under conditions similar to the drawing processes’. Thermal stability (DSC), X-ray photo analysis (XRD), and spectroscopic were measured. Eu^3+^ ions were used as spectral probes to determine the effect of P_2_O_5_ on the asymmetry ratio for the selected transitions (^5^D_0_ → ^7^F_1_ and ^5^D_0_ → ^7^F_2_). From the measurements, it was observed that the material produced exhibited amorphous or glass-ceramic properties, strongly dependent on the nucleator concentration. In addition, the conducted study confirmed that europium ions co-form the EuPO_4_ structure during the cooling process from 730 °C to room temperature. Moreover, the asymmetry ratio was changed from over 4 to under 1. The result obtained confirms that the developed material has properties typical of transparent glass-ceramic while maintaining high thermal stability, which will enable the fabrication of fibers with the glass-ceramic core.

## 1. Introduction

Glass-ceramics (GC) is a compound material with multiple application possibilities. After doping with trivalent lanthanide ions, it becomes an active glass-ceramic with many optoelectronic applications such as temperature sensing, lasers, and source of NIR, MIR, LED, displays, biomedical, and many others [1,2,3,4,5,6,7,8,9]. However, GC is usually used in bulk due to the technology gap in processing fibers. The lack of fiber manufacturing technology limits the application of active GCs in optical fibers; thus, the enormous potential for quantum efficiency, luminescence shaping, and other unique properties is mainly untapped [2,10,11]. Existing processing methods for glass-ceramic core optical fibers are complicated and make it impossible to obtain GC-core with a crystallization effect during the drawing process. The research in this paper leads to core materials that can crystallize during the drawing process without additional steps such as heat treatment. Rare earth (RE) ions also can be used as a spectral probe to detect structural changes [12,13]. The most used lanthanide in this role is the Eu^3+^ ion, which is used as a spectral probe to find the crystal phase of nanocomposite materials. According to the asymmetry ratio connected with 5D_0_ → 7F_1_ and ^5^D_0_ → ^7^F_2_ transitions, some structural changes in the vicinity of europium ions can be analyzed. The ^5^D_0_ → ^7^F_1_ transition is essentially independent of the structure of the matter around the europium ions and can be taken as a reference point in luminescence changes. In contrast, the 5D_0_ → ^7^F_2_ transition is of the hypersensitive type and strongly depends on the structure of the matter surrounding the europium ions [12,14,15,16].

Although the origins of transparent active GC doped with lanthanide ions date back to the 1970s [17] and optical fibers’ the 1960s [11,18], GC still poses problems when drawing optical fibers. Glass-ceramics compromise quantum efficiency [19,20,21], which depends on the local environment around RE ions, the chemical composition, luminescence, and thermal properties. For applications where the target product will be fibers, the crystallization strategy should focus on nano-ceramic materials with maximum crystal sizes below 20 nm–30 nm to maintain good transmission properties of the material and avoid scattering on large crystals [2,4,22]. The material’s thermal stability should also be kept in mind to obtain good quality fiber without unwanted crystallization connected with crystallization peaks [23,24]. The tested material meets many of the above postulates and has great potential for use as an optical fiber core. However, this may require further research and experimentation.

At present, there are several leading methods for the fabrication of active glass-ceramic materials. The first is direct doping involving the addition of crystals to the molten glass mass, and after quenching and annealing, manufactured material is a core material ready to be drawn in the rod-in-tube method. Unfortunately, the melting and, after that, the drawing process, degrades crystals [19,25]. Alternatively, the core material can be melted inside the cladding material, and this method is melt-in-tube (molten-core) or powder-in-tube [10,18,26,27]. This method of obtaining fibers involves diffusion problems between core and cladding materials, core irregularity and the difference between core and cladding in melting point, and viscosity [18]. The second classic method is based on manufacturing core bulk material. This material can have crystals or be grown in the heat treatment process [7,28,29]. Bulk material is the core material, but like in the first method, those crystals dissolve partially in the rod-in-tube drawing process. Alternatively, drawn optic fiber has an amorphous core, and crystals are grown in the heat treatment process after being drawn [10,18,26]. All these methods are characterized by the degradation of added crystals or problems with the amount and size of growth crystal, so more of these ways of obtaining optic fiber are multi-step and take a big amount of time on the heat treatment process [10,18,26]. An alternative to these methods should be the method based on obtaining the glass-ceramic structure in the process of fiber drawing.

Based on our earlier investigations, SGS glass was chosen for this investigation because of its good thermal stability, and low phonon energy glass matrix, which can improve quantum efficiency and probability of radiative transitions for electrons of rare-earth ion dopant [4,6,30]. Another reason was the possibility of shaping SGS into optic fibers. This research analyzed the effect of P_2_O_5_ concentration on the luminescence spectra, asymmetry ratio (AR), and luminescence decay. Due to earlier work [30], we decided to limit the P_2_O_5_ concentration up to 1 mol%, and we mainly focused on investigating the crystallization process in antimony-germanate-silicate glass, enabling size and density control of EuPO_4_ nanocrystals. The main objective of this experiment was to obtain a good optical quality core material as an 11 mm dia rod with specific crystalline properties and crystallization conditions.

## 2. Materials and Methods

Glasses with chemical composition (molar) 35SiO_2_ − 5Al_2_O_3_ − (29.5 − x)GeO_2_ − 20Sb_2_O_3_ − 10Na_2_O − xP_2_O_5_, (SGS) doped with 0.5Eu_2_O_3_, where x = 0, 0.25, 0.5, 0.75 and 1, were prepared by the melting-quenching method. The set was prepared from Sigma-Aldrich materials of >99.99% purity. The components were ground and homogenized in an agate mortar, and melted in an alumina crucible at 1450 °C by 2 h. Next, the molten material was poured into a brass mold to form 11 mm diameter rods. The bars were annealed at 20 °C below T_g_ for 8 h. In the next step, the rod was cut into 3 mm discs, which were polished to a final thickness of 2.5 mm. Some of the samples were subjected to the heat process, Samples were placed in a furnace preheated up to 730 °C. After the furnace temperature stabilized at 730 °C again, heating was turned off, and the furnace cooled down to room temperature in 7 h.

DSC measurement in the temperature range from 200 to 1000 °C was performed at 10 °C/min using the SETARAM Labsys thermal analyzer (Setaram Instrumentation, Caluire, France). X-ray diffraction studies were carried out on the X’Pert Pro X-ray diffractometer supplied by PANalytical (Eindhoven, The Netherlands) with Cu Kα1 radiation (λ = 1.54056 Å) in the 2θ range of 5°–90°. The step size, time per step, and scan speed were as follows: 0.017°, 184.79, and 0.011°/s. The X-ray tube was operated at 40 kV and 40 mA, and a scintillation detector was used to measure the intensity of the scattered X-rays.

The excitation and luminescence spectra of the glasses in a range of 350–750 nm were measured using the JobinYvon Fluoromax4 spectrophotometer (Horiba Jobin Yvon, Longjumeau, France). A system PTI QuantaMaster QM40 (Horiba Instruments, New York, NY, USA) coupled with a tunable pulsed optical parametric oscillator (OPO), pumped by the third harmonic of the Nd:YAG laser (OpotekOpolette 355 LD, Carlsbad, CA, USA), was used for luminescence decay measurements. The laser system was equipped with a double 200 mm monochromator, a multimode UV-VIS PMT (R928), and Hamamatsu H10330B-75 detectors controlled by a computer. Luminescence decay curves were recorded and stored by a PTI ASOC-10 (USB-2500, Horiba Instruments, New York, NY, USA) oscilloscope with an accuracy of ±1 µs.

## 3. Results and Discussion

### 3.1. Differential Scanning Calorimetry

Figure 1 showed DSC curves where a small effect of phosphorous content on characteristic transformation temperature (T_g_) was observed. With the increasing P_2_O_5_ content, T_g_ slightly decreases as well as crystallization temperature T_x_ increases. There are only weak exothermic changes, which indicate a low tendency to crystallization. Fabricated SGS glasses are characterized by good thermal stability, and this is also the case here, for there are weak exothermic changes that indicate a low tendency to crystallization [31].

Analyzing reference glass without orthophosphate compound following formulas were used to calculate glass stability and crystals’ nucleation and growth [23,32]:(1)$$\Delta T={\;T}_{x}-T_{g}$$
(2)$$H_{r}=\frac{T_{x}-T_{g}}{T_{m}-T_{x}}\;$$
(3)$$T_{\mathrm{rg}}=\frac{T_{g}}{T_{m}}$$
(4)$$\;H^{\prime }\;=\frac{T_{x}-T_{g}}{T_{g}}$$
where ΔT reflects the glass’s devitrification tendency and has a factor from 117 °C for the sample without orthophosphate to 155 °C for a 1 mol% P_2_O_5_ sample. A thermal stability factor ΔT higher than 100 °C is a good indication for the possibility of drawing fibers from such a material [6,16,33]. Hruby’s parameter H_r_ is 0.28 for the reference sample and 0.44 for 1 mol% of the P_2_O_5_ sample and gives information on the nucleation rate of the crystals [34]. The range of T_x_–T_g_ is connected with a viscosity of melted material, and a wider gap means lower viscosity. Additionally, range T_m_–T_x_ is connected with crystal growth rate, for narrow gap crystal growth rate slows rapidly [34]. Simplified Hruby’s parameter H′ is 0.26 for the reference sample and 0.35 for a 1 mol% P_2_O_5_ sample, which confirms good thermal stability [16,23]. When T_rg_ is lower than 0.75 is a good mark of the homogenous crystal nucleation in volume, where T_rg_ > 0.75 could indicate surface crystallization [32]. In this case, for the undoped sample T_rg_ = 0.45 and for 1 mol%, the P_2_O_5_ is 0.59. In general, the DSC study demonstrated a slight decrease in T_g_ point with increasing nucleator concentration due to a lower phosphorous melting temperature. On the other hand, T_x_ increases, which can be related to the increasing tendency to crystallize the material.

### 3.2. X-ray Diffraction

X-ray diffraction research was performed to determine the structural changes of prepared glass and glass-ceramic. Table 1, based on patterns (Appendix A), summarizes the Sherrer formula estimates of the crystals obtained at several observed 2theta angles for the glasses before and after heat treatment. Based on the data analysis, it was observed that for samples doped with 0.25 mol% P_2_O_5,_ the first nanometric crystallites appeared after heat treatment. An increase of P_2_O_5_ content to 0.5 mol% deepens the degree of crystallization, resulting in new reflections at 2θ = 42° and 72°. In both samples, no crystallization was observed before heat treatment. Some surprise is the presence of large > 65 nm crystallites in the sample with 0.5 mol% phosphorous. Further increasing the P_2_O_5_ content up to 0.75 mol% results in the formation of crystallites in samples before heat treatment. A slight increase in crystallite size was observed around 2θ = 30° and more significant growth at 2θ = 32°. Material with 1 mol% nucleator doping resulted in large crystallites above 20 nm and the appearance of new nanocrystallites for 26°.

For all cases where crystals were present, they were identified as monoclinic EuPO_4_ crystals (reference code 00-025-1055). In the EuPO_4_ crystal lattice, europium ions are coordinated with Eu-O and P-O atoms bonds forming polyhedrons [35,36]. In our experiment, the bulk crystallization mechanism was observed [37,38].

### 3.3. Excitation Spectra

Excitation spectra studies at two wavelengths, 590 nm and 612 nm, were performed to determine the effect of the molar content of nucleator ions and the heat treatment process on the excitation bands as a function of P_2_O_5_ concentration. The excitation spectra of the fabricated samples were measured for emission monitored at the wavelengths of 590 nm (Figure 2) and 612 nm (Figure 3), corresponding to magnetic dipole and electric-dipole transitions, respectively. In both cases, seven bands were observed on the examination spectrum at wavelengths: 360, 380, 395, 415, 465, 528, and 535 nm. Specific excitation bands may be combined with a magnetic (MD) or electric dipole (ED). Moreover, they can be connected to hypersensitive transitions. In this case, the most interesting transitions are described as hypersensitive and environment independent. Table 2 presents the chosen transition to analyzing examined samples [14].

Figure 2 and Figure 3 show excitation spectra normalized by ^7^F_0_ → ^5^D_1_ transition, which is environment independent, making it possible to observe changes in the most interfering excitation bands, i.e., 395 nm and 465 nm.

According to excitation spectra analysis monitoring at 590 nm, we observed that the highest intensity band and the hypersensitive bands show some rules, with hypersensitive transitions (^7^F_0_ → ^5^D_2_ and ^7^F_1_ → ^5^D_1_) decreasing their intensity with orthophosphate concentration, and this effect has a slight change between the heat treatment sample and before the heat treatment sample. Therefore, with more, up to 0.5 mol% P_2_O_5_ concentration, there are almost no changes in intensity compared to the reference sample (without P_2_O_5_), but with 0.75 mol% of the P_2_O_5_ threshold intensity declining rapidly. A different situation is with the ^7^F_0_ → ^5^L_6_ transition, their intensity increases with the concentration raise of orthophosphate. The 395 nm band changes after heat treatment, low doped by P_2_O_5_ samples strongly decreasing in intensity, contrary to the 0.75 mol% and 1.0 mol% doped samples decreasing slightly.

In the case of excitation spectra monitored at 612 nm, the analysis shows similar behavior to spectra monitored at 590 nm but with some differences in intensity linked with the heat treatment process. Therefore, more interesting maximum intensities of 390–405 nm and 460–470 nm bands were almost unchanged after heat treatment. This effect and changes for the 395 and 465 nm band confirm that the europium ions are in the crystalline phase. Surprisingly, there was a strong change in the intensity for the ^7^F_0_ → ^5^D_1_ transition in the annealed samples. This transition, similarly to the hypersensitive ^7^F_0_ → ^5^D_2_ transition, is strongly dependent on environment structure. The study confirmed the change in the structure of the samples, both as a function of annealing and nucleator content.

### 3.4. Luminescence Investigation and Decay

According to the results, we decided to use both excitation bands for further study on luminescence and decay. The main objective of the luminescence studies was to confirm that the luminescence changes depend on the P_2_O_5_ concentration, which is a confirmation of the incorporation of europium ions into the crystal structure. Similar to excitation spectra, there are also hypersensitive and independent of environment transitions. Selected information about that phenomena are in Table 3 [12,14,39,40,41].

The ^5^D_0_ → ^7^F_1_ transition is observed to be magnetic dipole dependent while, at the same time, the transition is independent of the structure of the matter surrounding the europium ions. Therefore, the luminescence in this band was normalized and used as a reference. In contrast, the ^5^D_0_ → ^7^F_2_ transition depends on the structure of matter around the europium ions. By comparing the luminescence for the ^5^D_0_ → ^7^F_1_ and ^5^D_0_ → ^7^F_2_ transitions, the asymmetry factor can be determined, which allows us to assume the presence of lanthanide ions in the crystal lattice. For this purpose, the incident results were processed using the formula [12,16,42]:(5)asymmerty ratio=∫604nm632nmD50→F72∫582nm604nmD50→F71

The study was conducted for two selected excitation wavelengths: 395 nm (Figure 4) and 465 nm (Figure 5) before and after the heat treatment process.

In the first analyzed case, it was observed that for excitation at 395 nm, the ^5^D_0_ → ^7^F_2_ transition strongly depends on the concentration of P_2_O_5_. The phosphorus oxide content up to 0.5 mol% hardly changes the luminescence compared to the sample without phosphorus oxide, but higher concentrations strongly change the luminescence spectrum, especially the ^5^D_0_ → ^7^F_2_ transition. Due to the 612 nm band being hypersensitive and strongly depending on the environment, such a change is a strong signal, indicating structural changes around Eu^3+^ ions. After the heat-treatment process, the luminescence changes are even more profound. There are interesting changes for the ^5^D_0_ → ^7^F_1_ transition, where a change of active sublevels (Stark’s effect) can be observed and resulting from changes in the structure of matter surrounding the europium ions. Usually, ^5^D_0_ → ^7^F_1_ splits into three Stark’s sublevels when the ion is located in a low symmetry site [12], and this situation is observed for concentrations up to 0.5 mol% P_2_O_5_. For higher concentrations of phosphorus oxide, ^5^D_0_ → ^7^F_1_ splits into two sublevels. Changes in luminescence in the test samples are translated into changes in the asymmetry ratio. Between phosphorus oxide concentrations are 0.5 mol% and 0.75 mol% the asymmetry ratio (AR) decline rapidly.

The luminescence under 464 nm has the highest intensity and the most dominant ED transition (Figure 5). The different emission profiles propose that the site-selective luminescence is possible in fabricated glass-ceramic. The emission spectra confirm the occupancy of Eu^3+^ ions at different crystallographic environments with varying asymmetric ratios. To better understand the mechanisms of crystallization of EuPO_4_ nanocrystals, the luminescence decay at two excitation channels are analyzed in detail. Figure 6 and Figure 7 show that glasses without or 0.25 mol% P_2_O_5_ content are purely amorphous. However, fabricated glasses where P_2_O_5_ content is higher than 0.5 mol% show a composite structure, which indicates the presence of crystals. This is particularly evident in samples with 0.75 mol% and 1.0 mol% phosphorus content. For amorphous materials, the results are approximated by a single-exponential function, while the appearance of crystalline structures requires a dual-exponential function [4,43]. Data from the experiment were fitted by the following equation:(6)$$I\left(t\right)=A_{1}\exp\left(-\frac{t}{\tau_{1}}\right)+A_{2}\exp\left(\frac{t}{\tau_{2}}\right)$$

To more easily compare lifetimes of different samples, the averaged observed lifetimes were calculated with the formula:(7)$$\tau_{\mathrm{avg}}=\frac{A_{1}\tau_{1}^{2}+A_{2}\tau_{2}^{2}}{A_{1}\tau_{1}+A_{2}\tau_{2}}$$
where τ_1_ and τ_2_ are long and short luminescence lifetime components, and A_1_ and A_2_ are constants amplitudes of respective decay components. The double-exponential character of the decay curves indicates the presence of two different surroundings of europium ions. Long lifetime τ_1_ corresponds to higher symmetry of the crystal field, while short lifetime τ_2_ is connected with lower symmetry of the Eu^3+^ surroundings [4,12,43].

As in previous studies, this study also confirmed a change in the structure of the matter surrounding the Eu^3+^ ions. Table 4 and Table 5 show that two concentrations are critical 0.5 mol% and 0.75 mol% P_2_O_5_. Excitation at 395 nm shows that sample P_0.5_ is the first double exponential, and τ_1_ changes rapidly because of a new relaxation channel. For 0.5 mol% P_2_O_5_ concentrations and higher, τ_1_ grows slowly, and τ_2_ decreases. After heat treatment, τ_2_ changes fast, and it is almost flattered for P_0.75_ and P_1.0_ samples.

Excitation at 465 nm before heat treatment is analogous to excitation at 395 nm before heat treatment. A different situation is that after heat treatment, a known edge at 395 nm for 0.5 mol% P_2_O_5_ concentration moves to 0.75 mol% P_2_O_5_ concentration. Flattered zones for 0.75 mol% and 1.0 mol% concentrations of P_2_O_5_ are informative about the maximum concentration of P_2_O_5_. Therefore, higher concentrations of orthophosphoric will not significantly affect the shortened lifetime.

## 4. Conclusions

In this study, we determined the mechanism of crystallization of EuPO_4_ nanocrystals located in antimony-germanate-silicate glass. Luminescence and excitation studies indicate that samples with 0.5 mol% P_2_O_5_ and higher concentration are characterized by a lower asymmetry ratio in the excitation scheme at 394 nm. It was noticed that the local symmetry exists in the vicinity of Eu^3+^ ions in this sample. Based on XRD measurements, nanocrystals with sizes in the range of 20–40 nm were found in almost all the glass samples tested. Moreover, the lifetime studies confirm the critical concentration of P_2_O_5_, where we can clearly see a sharp decrease in lifetimes for concentrations of 0.5–0.75 mol% P_2_O_5_. It is also easy to see that the dynamics of change is much higher for 394 nm excitation and for annealed samples. The rapid change in τ_2_ also indicates energy dissipation on large crystals, which behave as optical dimmers.

By proceeding with the investigation, the results were assumed that the material after melting should be amorphous, without crystals, while after heat treatment, the structure should change from amorphous to GC. The fabricated material meets many of the above postulates and has great potential for use as an optical fiber core, which is confirmed by thermal stability (DSC), X-ray analysis (XRD), and spectroscopic measurements. In the course of this study, it appears that SGS samples from a 0.25 mol% to 0.75 mol% nucleator concentration seem to be the most interesting in terms of the changes occurring in them and in terms of the strategy of obtaining nano-ceramic. However, in terms of the possibility of obtaining glass-ceramics in the form of optical fibers, the range of 0.25 mol% to 0.5 mol% P_2_O_5_ seems to be a better choice in terms of the dynamics of changes and thus the stability of the one-step process under development. As mentioned in the introduction, glass-ceramics are still under challenge, hence the fabricated antimony-germanate-silicate glass-ceramic is a compromise between luminescent properties and fiber drawing ability.

## Figures and Tables

**Figure 1 materials-15-03797-f001:**
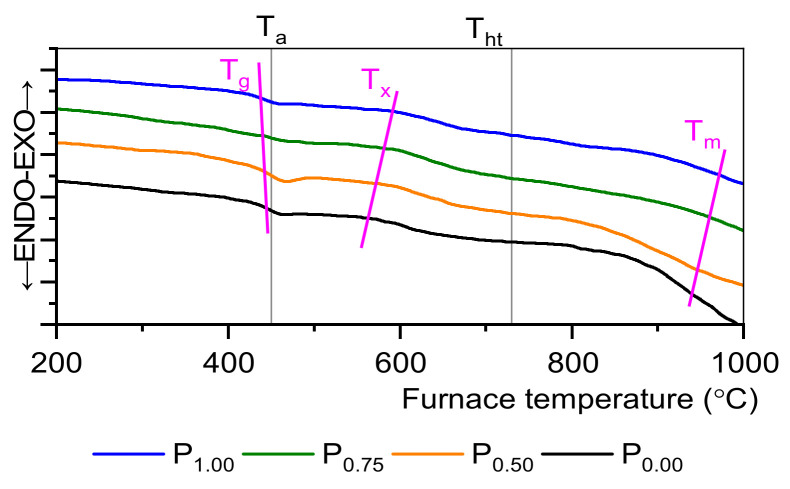
DSC curves. Temperatures: T_g_—glass transition, T_ht_—heat treatment, T_x_—crystallization, T_a_—annealing, and T_c_—crystallization peak.

**Figure 2 materials-15-03797-f002:**
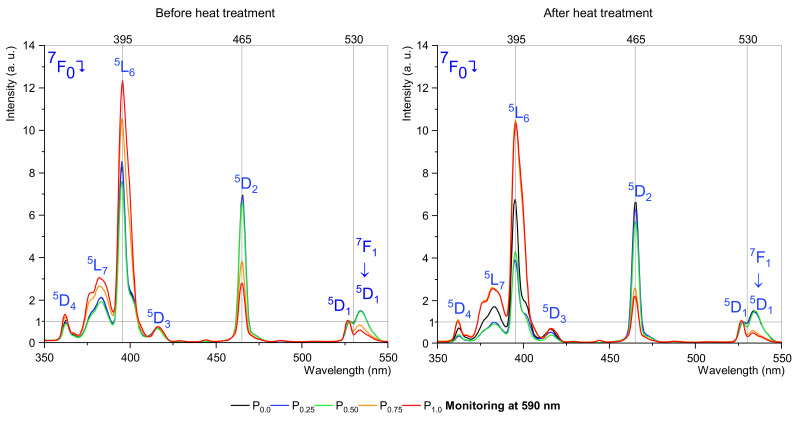
Excitation spectra, monitoring at 590 nm.

**Figure 3 materials-15-03797-f003:**
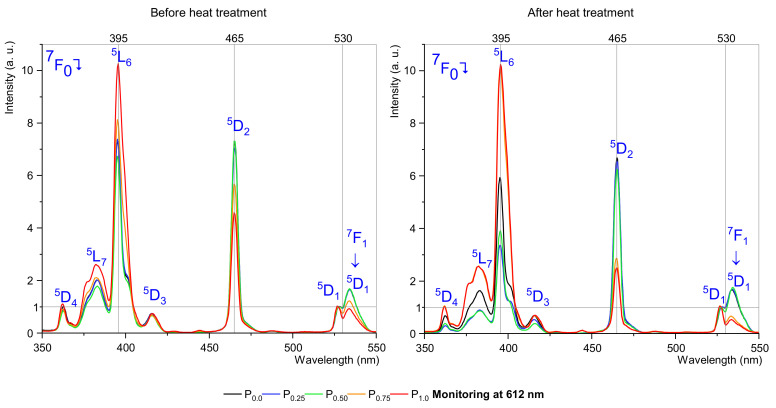
Excitation spectra, monitoring at 612 nm.

**Figure 4 materials-15-03797-f004:**
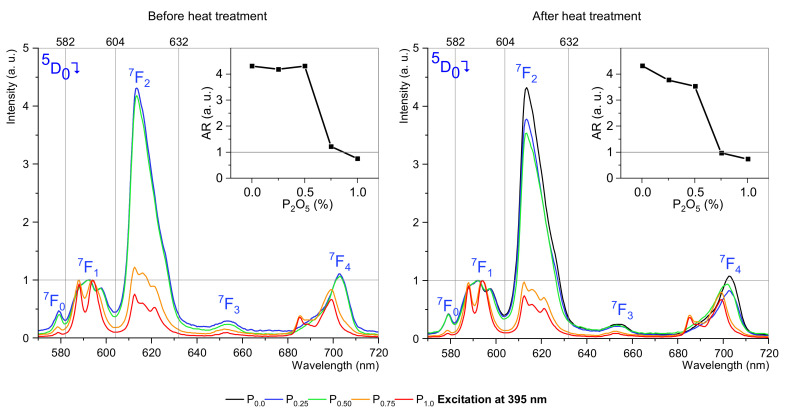
Luminescence spectra of SGS glasses doped with 0.5 mol%. In insets, asymmetry ratios are as a function of P_2_O_5_ concentration.

**Figure 5 materials-15-03797-f005:**
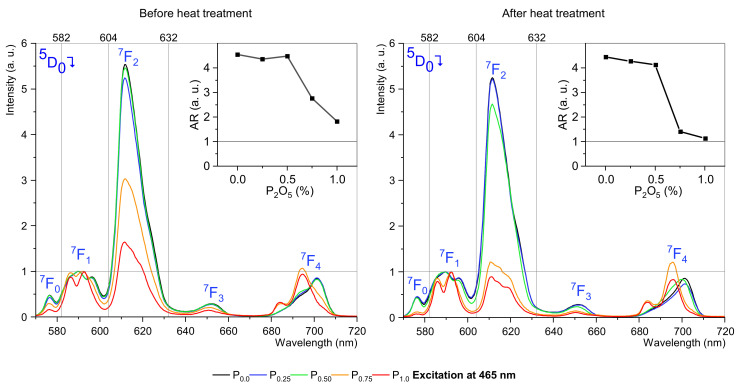
Luminescence spectra of SGS glasses doped with 0.5 mol% Eu_2_O_3_. In insets, asymmetry ratios as a function of P_2_O_5_ concentration.

**Figure 6 materials-15-03797-f006:**
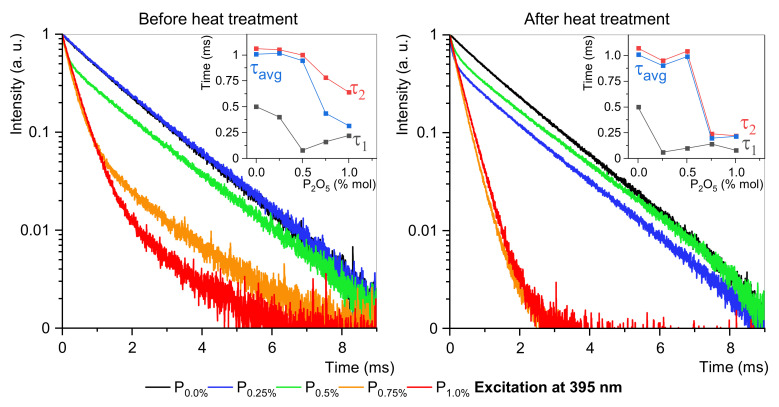
Luminescence decays and τ changes at 395 nm excitation. Lifetimes are in insets.

**Figure 7 materials-15-03797-f007:**
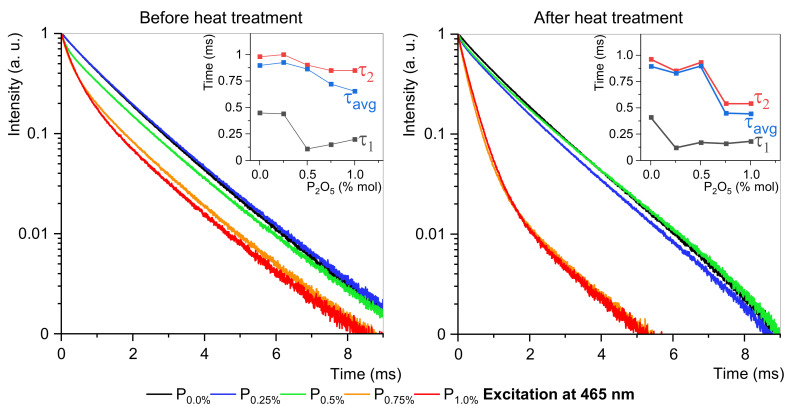
Luminescence decays and τ changes at 465 nm excitation. Lifetimes are in insets.

**Table 1 materials-15-03797-t001:** XRD table crystallite size depends on theta angle, nucleator concentration, and heat treatment process (all crystallite sizes in nm).

P_2_O_5_ Nucleator Concentration	0.25 mol%		0.50 mol%		0.75 mol%		1.0 mol%	
2θ	Bht ^1^	Aht ^2^	Bht	Aht	Bht	Aht	Bht	Aht
26								5.82
28							20.66	8.26
30		12.53		16.27	14.28	15.89	22.75	30.95
32		11.49		15.85	16.92	27.24		43.93
42				65.50				
72				28.18				

^1^ Bht—Before heat treatment; ^2^ Aht—After heat treatment.

**Table 2 materials-15-03797-t002:** The chosen transitions of Eu^3+^ ions.

Transition	Dipole Type	Wavelength Range (nm)	Transition Describes
^7^F_0_ → ^5^L_6_	ED	390–405	Most intense transition
^7^F_0_ → ^5^D_2_	ED	460–470	Hypersensitive transition
^7^F_0_ → ^5^D_1_	MD	520–530	Intensity independent of the environment
^7^F_1_ → ^5^D_1_	ED	530–540	Hypersensitive transition

**Table 3 materials-15-03797-t003:** The selected radiative transitions of Eu^3+^ ions.

Transition	Dipole Type	Wavelength Range (nm)	Transition Describes
^5^D_0_ → ^7^F_1_	MD	585–600	Intensity largely independent of the environment
^5^D_0_ → ^7^F_2_	ED	610–630	Hypersensitive
^5^D_0_ → ^7^F_4_	ED	680–710	Sensitive, depend on environment

**Table 4 materials-15-03797-t004:** Lifetime of ^5^D_0_ level at excitation of 395 nm.

Sample	Before Heat Treatment, Excited at 395 nm						After Heat Treatment, Excited at 395 nm					
	A_1_	τ_1 [ms]_	A_2_	τ_2 [ms]_	τ_avg_. _[ms]_	R^2^	A_1_	τ_1 [ms]_	A_2_	τ_2 [ms]_	τ_avg [ms]_	R^2^
P_0.0_	0.97	0.97			0.97	0.9998	0.99	0.94			0.94	0.9996
P_0.25_	0.98	0.96			0.96	0.9997	0.63	0.72			0.72	0.9703
P_0.5_	0.44	0.08	0.54	1.00	0.9437	0.9995	0.37	0.10	0.63	1.04	0.9897	0.9997
P_0.75_	0.85	0.16	0.14	0.78	0.7361	0.9996	0.58	0.14	0.43	0.24	0.1959	0.9999
P_1.0_	0.89	0.22	0.09	0.64	0.3154	0.9998	0.11	0.08	0.9	0.22	0.2140	0.9998

**Table 5 materials-15-03797-t005:** Lifetime of ^5^D_0_ level at excitation of 465 nm.

Sample	Before Heat Treatment, Excited at 465 nm						After Heat Treatment, Excited at 465 nm					
	A_1_	τ_1 [ms]_	A_2_	τ_2 [ms]_	τ_avg_. _[ms]_	R^2^	A_1_	τ_1 [ms]_	A_2_	τ_2 [ms]_	τ_avg [ms]_	R^2^
P_0.0_	0.98	0.83			0.83	0.9993	0.98	0.82			0.82	0.9994
P_0.25_	0.97	0.86			0.86	0.9993	0.89	0.76			0.76	0.9975
P_0.5_	0.28	0.11	0.71	0.90	0.8636	0.9998	0.91	0.81			0.81	0.9974
P_0.75_	0.55	0.15	0.43	0.85	0.7210	0.9998	0.82	0.16	0.77	0.54	0.4488	0.9998
P_1.0_	0.64	0.20	0.35	0.85	0.6455	0.9998	0.86	0.18	0.78	0.54	0.4432	0.9999

## Data Availability

Not applicable.

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
