# Peer review of "Crystallization Mechanism and Optical Properties of Antimony-Germanate-Silicate Glass-Ceramic Doped with Europium Ions"

_materials, 2022, doi:10.3390/ma15113797_

Round 1

Reviewer 1 Report

In this paper, the authors study the influence of the variation on the P2O5 content, in the process of obtaining some materials such as ceramic glass, on the specific properties. After analyzing the manuscript, I recommend publishing it, but I would like to make some recommendations

  1. The degree of research innovation must be very well highlighted
  2. Line 38, to be reformulated, is repeated as in Abstract
  3. The text between lines 86 and 91 should be moved to the results and discussion section
  4. It is necessary for the authors to present clearly in the Materials and methods section, what chemicals they used in the research and what is their source, concentrations, etc.
  5. Line 103, the phrase about the X-ray analysis should be entered in line so that the information can be tracked more easily.
  6. Lines 162-163 the expression in English is not ok, it needs to be reformulated.
  7. Lines 179, 191, please remove monitoring
  8. Regarding the expression of P2O5 content, several units of measurement appear in the text, for example: 0.5%, 0,5mol%, 0,5%mol, please review to be consistent.
  9. Line 289, please change "approx.". I don't think it's ok to have abbreviations in this context, if they aren't defined.
  10. I recommend a rephrase of the conclusions, in order to be clear and concise, so that the scientific value of the research can be easily followed.

Author Response

Dear Reviewer 1,

thank you for your valuable remarks. Please find our answers attached in file.

regards, JZ

Reviewer 2 Report

The study presents the effects of the phosphorous on the crystallization, thermal stability, and optical properties. It shows that with a proper level of phosphorous the material can be able to possess properties character of transparent glass-ceramics, meanwhile to maintain a high thermal stability necessary for fabricating fibers with a glass-ceramic core. To be honest, I am not specialist for optical properties, no comments would be given for this part. Regarding to the sample fabrication and characterization, maybe the authors consider:

1, more details on the heat process: how fast to warm up and to cool down the samples, and how long to dwell at 730°C (730 or 700°C, inconsistent in the paper).

2, details on the estimation of the crystallites size: how to substrate the contribution from the instrument to the FWHM? How reliable of the estimation in such small peaks? TEM for confirmation?

3, explicit effects of the phosphorous on Tg

4, definition of the symbols in Eqs. (Tx etc.)

5, explicit description on the mechanism of crystallization of EuPO4.

6, grammar and format checking. 

Author Response

Dear Reviewer 2,

thank you for your valuable remarks. Please find our answers attached in file.

regards, JZ

Reviewer 3 Report

The manuscript describes the properties of glass-ceramics to be used in fibers by doping with phosphate and europium ions. In general, the glasses are well-characterized, but I have the following suggestions for improving the manuscript.

The abstract misses to mention the type of glass used for the research.

In the introduction, the authors should describe why it is of importance to make glass-ceramic fibers.

Please describe why the antimony-germanate-silicate glass is a composition of interest.

Please state in line 99, why the glasses are annealed at 450 C. What is this temperature compared to the Tg?

The DSC method should be improved. The Tg measured by DSC depends on the previous cooling rate. The Tg should be measured with a heating rate of 10 C/min (as is done), but prior to that, the sample should have been cooled from above the Tg with 10 C/min as well.

In relation to Fig. 1, the exact glass transition temperature should be stated for each glass composition, and it should be described why possible changes occur. Also, as Tm is used in equations 2 and 4, I suggest to show that in Fig. 1 like Tg and Tht.

Regarding the XRD diffractograms in Supporting Information. What is the explanation of the two additional peaks for 0.5% P2O5? Also, the significance of intensity of the new peak at 42 indicates something else is crystallized as well, as the intensity overcomes the one of the peaks around 30 2theta which should be the most dominant peaks. I suggest to redo this sample.

Author Response

Dear Reviewer 3,

thank you for your valuable remarks. Please find our answers attached in the file.

regards, JZ

Round 2

Reviewer 2 Report

To me the paper is ready for publication. 

Reviewer 3 Report

The authors corrected all concerns. It can now be published.